# *Bmal1* in the striatum influences alcohol intake in a sexually dimorphic manner

Nuria de Zavalia [1✉], Konrad Schoettner [1], Jory A. Goldsmith[1], Pavel Solis[1], Sarah Ferraro[1], Gabrielle Parent[1] & Shimon Amir [1✉]

Alcohol consumption has been strongly associated with circadian clock gene expression in mammals. Analysis of clock genes revealed a potential role of *Bmal1* in the control of alcohol drinking behavior. However, a causal role of *Bmal1* and neural pathways through which it may influence alcohol intake have not yet been established. Here we show that selective ablation of *Bmal1* (Cre/loxP system) from medium spiny neurons of the striatum induces sexual dimorphic alterations in alcohol consumption in mice, resulting in augmentation of voluntary alcohol intake in males and repression of intake in females. *Per2*mRNA expression, quantified by qPCR, decreases in the striatum after the deletion of *Bmal1*. To address the possibility that the effect of striatal *Bmal1* deletion on alcohol intake and preference involves changes in the local expression of *Per2*, voluntary alcohol intake (two-bottle, free-choice paradigm) was studied in mice with a selective ablation of *Per2* from medium spiny neurons of the striatum. Striatal ablation of *Per2* increases voluntary alcohol intake in males but has no effect in females. Striatal *Bmal1* and *Per2* expression thus may contribute to the propensity to consume alcohol in a sex -specific manner in mice.

[1] Center for Studies in Behavioral Neurobiology, Department of Psychology, Concordia University, Montreal, Canada. ✉email: nuria.dezavalia@concordia.ca; shimon.amir@concordia.ca

**B**rain and muscle ARNT-like protein 1 (*Bmal1*) is a circadian clock gene and transcriptional regulator that plays an obligatory role in the generation of circadian rhythms in the suprachiasmatic nucleus (SCN), the master circadian clock in mammals[1]. Notably, *Bmal1* is widely expressed in the mammalian brain[2], and perturbations in *Bmal1* expression in distinct regions outside the SCN cause various physiological and behavioral disruptions[3–5], including disturbances in sleep architecture[6], and in cognitive and mood-related behaviors[7–12]. Over the past years, association analysis of clock genes revealed a potential role of *Bmal1* in the control of alcohol drinking behavior[13,14]. Kovanen and colleagues found evidence that *Bmal1* is linked to alcohol consumption[15]. Similarly, an association between *Bmal1* polymorphisms and an increased risk of alcohol

**Fig. 1 Alcohol drinking behavior of control and *Bmal1* knockout mice. a** Quantitative PCR analysis of *Bmal1* mRNA levels measured at ZT1 in dorsal striatal tissue of control, *Bmal1* heterozygote and knockout mice. ANOVA, significant genotype effect, $F_{(2, 7)} = 113.8$, $p < 0.0001$, Tukey's post-hoc test, *$p < 0.005$. **b** A representative image of BMAL1 immunofluorescence staining at ZT 1 in dorsal striatal tissue of control, *Bmal1* heterozygote and knockout mice. BMAL1: red, Gpr88-Cre-GFP: green. Scale bar = 30 μm. **c** Quantitative PCR analysis of *Per2* mRNA levels in dorsal striatal tissue of control, *Bmal1* heterozygote and knockout male mice. Two-way ANOVA, significant genotype effect, $F_{(2, 5)} = 21.46$, $p = 0.0035$, and time point effect, $F_{(2, 8)} = 51.7$, $p < 0.0001$, significant interaction effect, $F_{(4, 8)} = 13.97$, $p = 0.0011$. **d** Quantitative PCR analysis of *Dbp* mRNA levels in dorsal striatal tissue of control, *Bmal1* heterozygote and knockout mice. Two-way ANOVA, significant genotype effect, $F_{(2, 5)} = 116.6$, $p < 0.0001$, and time point effect $F_{(2, 8)} = 5.236$, $p < 0.05$ effect, Tukey's test, **$p < 0.05$. **e** Daily alcohol consumption (left) and average alcohol consumption (right) of control, and *Bmal1* knockout male mice. Two-way repeated measure ANOVA, (RM-ANOVA) significant genotype effect, $F_{(1, 23)} = 13.26$, $p = 0.0014$, Unpaired two-tailed t-test, **$p < 0.01$. **f** Daily alcohol consumption (left) and average alcohol consumption (right) of control, and *Bmal1* knockout female mice. RM-ANOVA, significant genotype effect, $F_{(1, 29)} = 5.0$, $p = 0.033$. Unpaired two-tailed t-test, *$p < 0.05$. **g** Daily alcohol preference (left) and average alcohol preference (right) of control, and *Bmal1* knockout male mice. RM-ANOVA, significant genotype effect, $F_{(1, 23)} = 10.73$, $p = 0.0033$. Unpaired two tailed t-test, **$p < 0.01$. **h** Daily alcohol preference (left) and average alcohol preference (right) of control, and *Bmal1* knockout female mice. RM-ANOVA, significant genotype effect, $F_{(1, 29)} = 4.708$, $p = 0.0384$. Unpaired two tailed t-test, *$p < 0.05$. **i** Daily fluid intake (left) and average fluid intake (right) of control and *Bmal1* knockout male mice. RM-ANOVA, no significant effect, $F_{(1, 23)} = 0.3737$, $p = 0.5470$. Unpaired two-tailed t-test, NS. **j** Daily fluid intake (left) and average fluid intake (right) of control and *Bmal1* knockout female mice. RM-ANOVA, no significant effect, $F_{(1,29)} = 0.3185$, $p = 0.5769$. Unpaired two-tailed t-test, NS. NS = no significant differences. CTR: control, HET: *Bmal1* heterozygote, SKO: *Bmal1* knockout. ZT: Zeitgeber time. **c-j**, the values express mean ± S.E.M. **a–d**, n = 3/genotype. **e**, **g**, **i**, CTR n = 12, SKO n = 13. **f**, **h**, **j**, CTR n = 17, SKO n = 14.

use disorder (AUD) has been described[16]. However, a causal role of *Bmal1* and neural pathways through which it may influence alcohol intake have not yet been established. The striatum, a subcortical part of the forebrain, is the major input site of the basal ganglia, a neuronal structure involved in the control of reward-related processes. Alcohol alters the function of striatal circuits in multiple ways, which may contribute to acute intoxication, alcohol-seeking, dependence and withdrawal[17–24]. The dorsal striatum is involved in the control of alcohol habit formation and goal-directed alcohol seeking, whereas the ventral striatum has an important role in environmental control of alcohol drinking and relapse[25].

Alcohol use, abuse and dependence are sex-dependent. In humans, females report lower alcohol use and dependency than males[26–29]. This pattern is reversed in rodents; females display higher alcohol intake compared to males[30–32]. In both humans and rodents, females suffer from more adverse consequences of alcohol use and dependency, spanning from physical health to cognition and mental health[26–29]. Interestingly, striatal function and morphology is sexual dimorphic, and affected by components of the circadian clock[33–39].

Therefore, we hypothesize that circadian clock genes affect ethanol consumption in mice in a sex dependent manner. We examine voluntary alcohol consumption and preference in male and female mice that lack *Bmal1* in medium spiny neurons (MSNs) of the striatum exclusively, which constitute approximately 95% of striatal neurons[25]. Our experiments reveal that *Bmal1* in MSNs exerts a sexually dimorphic influence on alcohol drinking behavior – repressing preference and intake in males and promoting high preference and intake in females – which may contribute to sex differences in the propensity to consume alcohol in mice. This mechanism seems to be mediated by *Per2* in male mice, whereas is appears to be independent of *Per2* in females.

## Results

### Generation and validation of striatal *Bmal1* knockout mice.
Conditional knockout mice that lack *Bmal1* exclusively in striatal MSNs were created by crossing mice that express floxed alleles of *Bmal1* with mice that express Cre recombinase under the control of *Gpr88*, a striatum specific G-protein coupled receptor. Knockout efficiency and specificity were validated by mRNA and protein expression analysis, confirming the striatum- specific absence of *Bmal1* and loss of circadian clock function in the striatum but not in the suprachiasmatic

nucleus (SCN), the master circadian clock. Specifically, quantitative-PCR analysis of striatal tissue revealed a substantial reduction of *Bmal1* mRNA levels in homozygote *Bmal1* knockout mice (Gpr88Cre/+; Bmal1fl/fl [Bmal1SKO]) and heterozygote knockout mice (Gpr88Cre/+; Bmal1fl/+ [Bmal1-HET]) compared to wildtype controls (Gpr88+/+; Bmal1fl/fl, [Bmal1CTR]) (Fig. 1a). Fluorescence immunohistochemistry shows that BMAL1 is expressed in striatal tissue sections from Bmal1CTR and Bmal1HET mice, whereas sections from Bmal1SKO mice lacked BMAL1 immunostaining (Fig. 1b). The reduction of BMAL1 in the striatum was further confirmed by western blotting analysis of striatal tissue at 2 different times across the 24 h day (Supplementary Figs. 1a, b, 3). In addition, we established that the deletion of *Bmal1* is restricted to the striatum by showing the presence of BMAL1 immunostaining in the SCN and hippocampus of knockout mice (Supplementary Fig. 1c). To determine if the deletion of *Bmal1* disrupted the striatal circadian clock, we studied the expression of the clock gene *Per2* and the canonical clock-controlled gene *Dbp*. In the striatum of Bmal1CTR mice, *Per2* mRNA levels peaked at night (Zeitgeber time (ZT) 17, i.e. 5 h after the light goes off). In contrast, in Bmal1SKO mice, striatal levels of *Per2* mRNA were significantly downregulated at ZT11 and ZT17, and its rhythm was blunted (Fig. 1c). Similarly, *Bmal1* deletion from MSNs induced a significant downregulation of *Dbp* mRNA in the striatum (Fig. 1d, Supplementary Data 1).

**Voluntary alcohol consumption in *Bmal1* knockout mice.** To study voluntary alcohol consumption, we used a two-bottle, free-choice paradigm. Mice had access to one bottle of 15% ethanol solution in tap water (vl/vl) and one bottle of tap water only, every other day, in alternate left-right position, for a total of 11 sessions. Analysis of variance revealed a significant main effect of genotype on alcohol intake and preference across the 11 test sessions in both males (Fig. 1e, g) and females (Fig. 1f, h). On average, Bmal1SKO males consumed 33% more alcohol and exhibited 36% greater alcohol preference than Bmal1CTR mice over the 11 alcohol test days. Strikingly, contrary to males, Bmal1SKO females consumed 22% less alcohol than Bmal1CTR females (Fig. 1f) and exhibited 15% lower alcohol preference (Fig. 1h, Supplementary Data 1) over the 11 alcohol test days. These results show that *Bmal1* in MSNs exerts a sexually dimorphic influence on alcohol drinking behavior in mice, repressing intake and preference in males and promoting them in females. Interestingly, in both

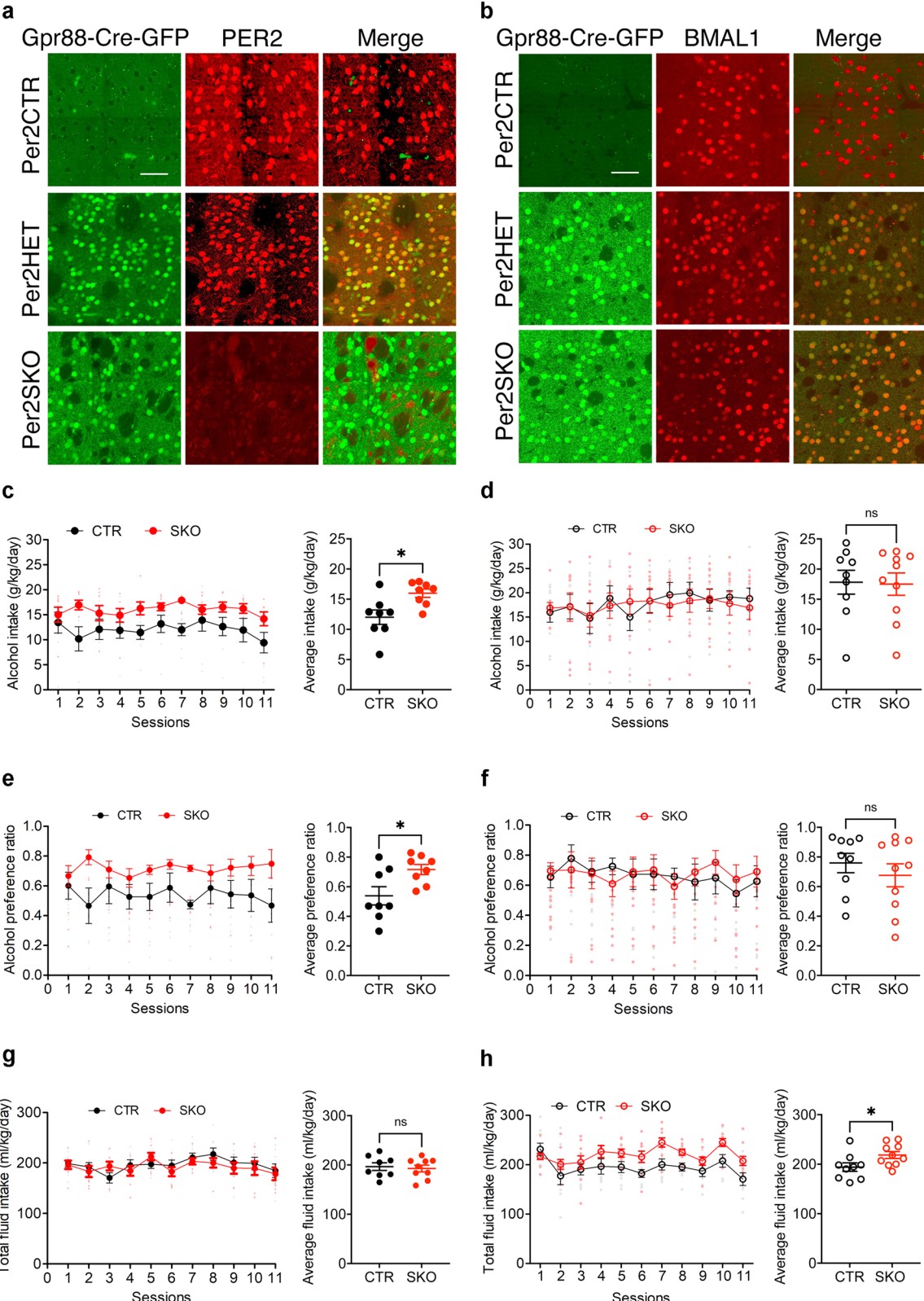

males and females, Bmal1HET show no difference in alcohol consumption compared to control animals (Fig. 3a–f and Supplementary Data 3), indicating that one copy of *Bmal1* in MSNs is sufficient to maintain the normal level of alcohol consumption. Total fluid intake (Fig. 1i, j) and body weight (Supplementary Fig. 1e–h, Supplementary Data 1) were comparable between genotypes in both sexes, establishing that the

changes in alcohol consumption in knockout mice are not the result of changes in body mass or general fluid intake.

**Voluntary alcohol consumption in *Per2* knockout mice.** *Bmal1* plays an obligatory role in the expression of the clock gene *Per2* in the striatum, as shown by gene expression analysis (Fig. 1c). *Per2* is associated with alcohol consumption in animal models and

**Fig. 2 Alcohol drinking behavior of control and *Per2* knockout mice. a** A representative image of PER2 immunofluorescence staining in dorsal striatal tissue of control and *Per2* knockout mice. PER2: red, Gpr88-Cre-GFP: green. $n = 3$/genotype, scale bar $= 50\,\mu$m. **b** A representative image of BMAL1 immunofluorescence staining in dorsal striatal tissue of control, and *Per2* knockout mice. BMAL1: red, Gpr88-Cre-GFP: green. $n = 3$/genotype, scale bar $= 50\,\mu$m. **c** Daily alcohol consumption (left) and average alcohol consumption (right) of control and *Per2* knockout male mice. Two-way repeated measure ANOVA (RM-ANOVA), significant genotype effect, $F_{(1, 14)} = 8.332$, $p = 0.0120$. Unpaired two-tailed t-test, *$p < 0.05$. **d** Daily alcohol consumption (left) and average alcohol consumption (right) of control and *Per2* knockout female mice. RM-ANOVA, no significant effect, $F_{(1, 17)} = 0.014$, $p = 0.9072$. Unpaired two-tailed t-test, NS. **e** Daily alcohol preference (left) and average alcohol preference (right) of control and *Per2* knockout male mice. RM-ANOVA, significant genotype effect, $F_{(1, 14)} = 6.552$, $p = 0.0227$. Unpaired two-tailed t-test, *$p < 0.05$. **f** Daily alcohol preference (left) and average alcohol preference (right) of control and *Per2* knockout female mice. RM-ANOVA, no significant effect, $F_{(1, 17)} = 0.01779$, $p = 0.8955$. Unpaired two-tailed t-test, NS. **g** Daily fluid intake (left) and average fluid intake (right) of control and *Per2* knockout male mice. RM-ANOVA, no significant effect, $F_{(1, 14)} = 0.142$, $p = 0.7116$. Unpaired two-tailed t-test, NS. **h** Daily fluid intake (left) and average fluid intake (right) of control and *Per2* knockout female mice. RM-ANOVA, significant genotype effect, $F_{(1, 17)} = 5.665$, $p = 0.0293$. Unpaired two-tailed t-test, *$p < 0.05$. NS $=$ no significant differences. CTR: control, HET: *Per2* heterozygote, SKO: *Per2* knockout. **c–h** The values express mean ± S.E.M. **a**, **b** $n = 3$/genotype. **c**, **e**, **g**, CTR $n = 8$, SKO $n = 8$. **d**, **f**, **h** CTR $n = 9$, SKO $n = 10$.

humans[40]. Global loss of function of *Per2* has been shown to augment alcohol intake and preference in male mice[40], raising the possibility that the effect of striatal *Bmal1* deletion on alcohol intake and preference involve changes in the local expression of *Per2*. To study the direct contribution of striatal *Per2* expression in alcohol drinking, we generated striatum-specific *Per2* knockout male and female mice by crossing animals carrying floxed alleles of *Per2* with mice expressing Cre recombinase under the control of the *Gpr88* promoter. Immunostaining of brain sections from Per2SKO mice revealed a complete absence of PER2 immunoreactivity in the striatum (Fig. 2a), but not in the cortex (Supplementary Fig. 1d). The deletion of *Per2* from the MSNs augmented voluntary alcohol intake and preference in male mice (Fig. 2c, e), thus mimicking the effect of striatal *Bmal1* deletion. On average, daily alcohol intake and preference of Per2SKO male mice were 33% higher than control littermates. Importantly, the absence of *Per2* expression in the MSNs did not affect *Bmal1* expression (Fig. 2b, Supplementary Data 2), indicating that the effect of *Per2* deletion on alcohol consumption is independent of *Bmal1*.

In contrast to males, the removal of *Per2* from the MSNs did not affect alcohol consumption and preference in females (Fig. 2d, f), revealing a female-specific dissociation between the effect of *Bmal1* and *Per2* on alcohol intake. These results suggest that the effect of *Bmal1* deletion on alcohol behavior could be mediated by *Per2* in males. On the contrary, this effect does not rely on *Per2* and is directly related to the loss of function of *Bmal1* in the striatum in females.

Deletion of one copy of *Per2* (Per2HET) had no effect on alcohol intake and preference in males or females, indicating that striatal *Per2* is haplosufficient in the control of alcohol intake (Fig. 3g–l, Supplementary Data 3). Finally, similar to our findings for the deletion of *Bmal1*, no difference in body weight (Supplementary Fig. 1i–l) was found across the 11 sessions. The total liquid consumption was significantly different for the Per2SKO females compared to the controls (Fig. 2h, Supplementary Data 2).

**Sucrose consumption in *Bmal1* and *Per2* knockout mice.** Alcohol consumption and preference have been associated with the propensity to consume sweet solutions[41]. To study whether the sexually dimorphic effects of striatal deletion of *Bmal1* or *Per2* on alcohol behavior extend to, or may have resulted from, a change in the general sensitivity to palatable taste, we assessed sucrose consumption in Bmal1SKO and Per2SKO mice.

We found that the striatal deletion of *Bmal1* or *Per2* did not affect voluntary intake of 0.25% or 2.0% sucrose solution, delivered in a two-bottle free-choice paradigm with water for 3

consecutive days, in males (Fig. 4a, c, e, g) or females (Fig. 4b, d, f, h, Supplementary Data 4).

Thus, the effects of striatal deletion of *Bmal1* or *Per2* from MSNs on alcohol intake and preference appears to be specific and not due to changes in the palatability of alcohol taste.

**Circadian rhythms analysis in *Bmal1* and *Per2* knockout mice.** Global or selective deletion of *Bmal1* or *Per2* in the SCN disrupts circadian behavioral rhythms[1,42], and disruption of circadian rhythms can influence alcohol consumption[43,44]. However, both types of rhythm disruptions comprise systemic alterations of circadian functions and do not allow to draw conclusions on the exact origin of the observed behavioral phenotypes. To ensure that the differences in alcohol consumption observed in striatal *Bmal1* and *Per2* knockout animals were not due to indirect effects of the conditional knockout on SCN function, we monitored wheel- running behavior in alcohol naïve mice housed individually under different lighting conditions.

Circadian activity rhythms in Bmal1SKO, Bmal1HET, Per2SKO and Per2HET male and female mice were indistinguishable from those in respective male and female controls (Fig. 5). In particular, striatal deletion of two (SKO) or only one copy (HET) of *Bmal1* or *Per2* did not affect activity levels and entrainment of daily rhythms of wheel running to a 12:12 h light-dark cycle (Figure 5a–f), adjustments to phase shifts in the light cycle (Fig. 5g–n), and free running in constant conditions (Fig. 5o–v, Supplmentary Data 5). These results show that Bmal1SKO and Per2SKO male and female mice and their respective HET mice have a functional SCN and normal circadian pacemaking. Thus, the sexually dimorphic changes in alcohol consumption in SKO and HET male and female mice are specific to the clock gene manipulation in MSNs of the striatum.

**Alcohol consumption in *Gpr88* +/cre mice.** Bmal1SKO and Per2SKO and the respective HET genotypes have only one functional copy of *Gpr88* in MSNs, since one copy is modified to drive *Cre* and EGFP expression. Complete deletion of *Gpr88*, has been shown to augment alcohol intake in male mice[45], raising the possibility that the sexually dimorphic changes in alcohol consumption seen in striatal Bmal1SKO and Per2SKO were due, at least in part, to a sex difference in a contributory effect of *Gpr88* monoallelic expression in MSNs. To study this possibility, we compared alcohol intake and preference between *Gpr88*+/+ mice and *Gpr88*Cre/+ mice. Average alcohol intake and preference in *Gpr88*Cre/+ male and female mice were similar compared to *Gpr88* +/+ controls (Fig. 6, Supplementary Data 6), excluding the possibility that the changes in alcohol consumption associated with deletion of striatal *Bmal1* or *Per2*

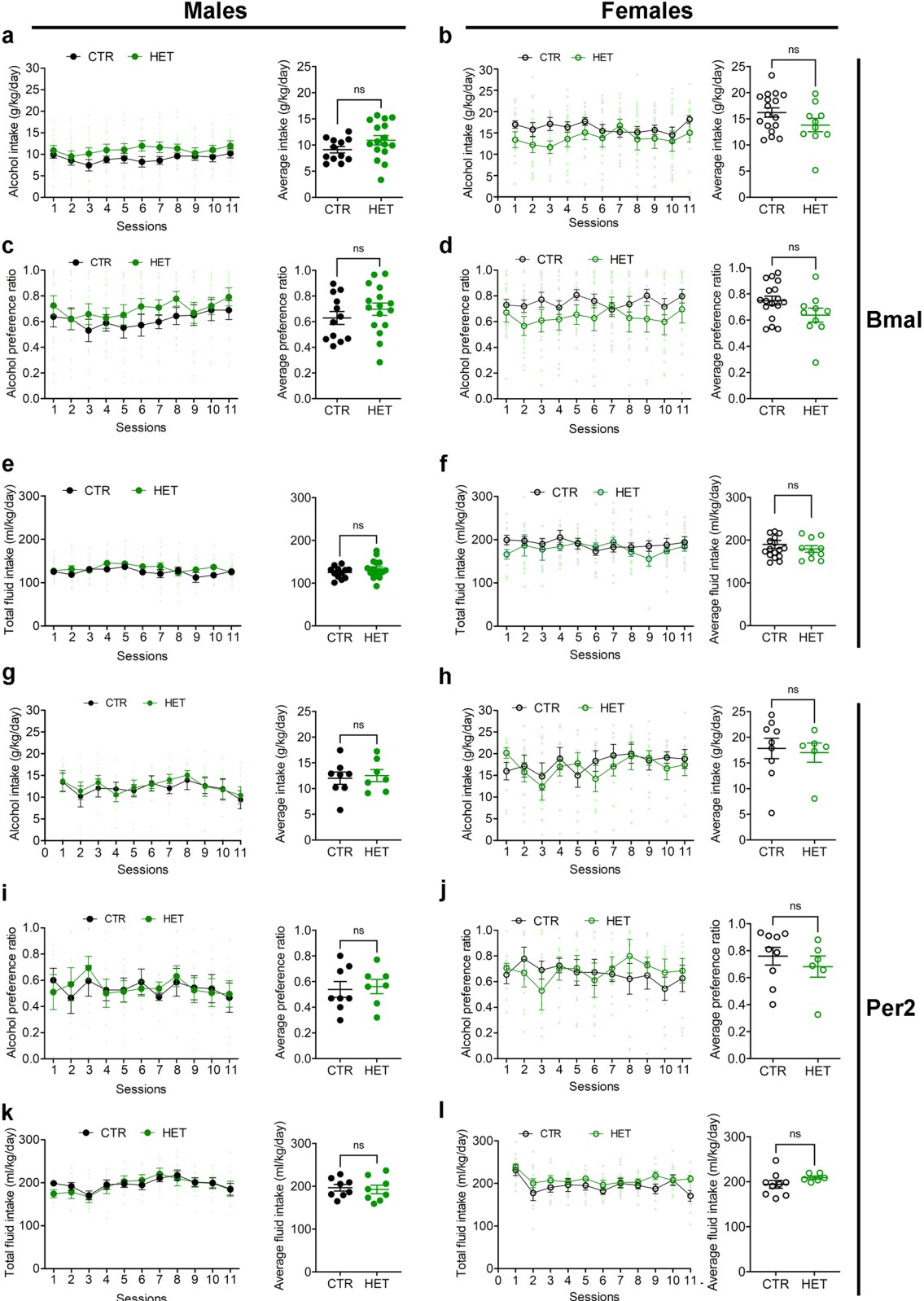

were due to *Gpr88* haploinsufficiency, or expression of Cre-EGFP in *Gpr88*- bearing cells[46].

Sex differences in alcohol behavior are well recognized in rodents, with females generally drinking more alcohol than males[33–35]. Consistent with this, average daily alcohol intake and preference in female control mice was significantly greater than in males (Supplementary Fig. 2a, d). In contrast, mean alcohol preference in Bmal1SKO male mice was significantly higher than Bmal1SKO females, and no difference in mean alcohol intake was observed (Supplementary Fig. 2c, f). Similarly, average alcohol intake and preference were greater in Per2CTR females than Per2CTR males (Supplementary Fig. 2g, j), whereas no significant differences in mean alcohol preference and intake were noted between Per2SKO males and females (Supplementary Fig. 2i, l).

**Fig. 3 Alcohol drinking behavior of control, *Bmal1*, and *Per2* heterozygote mice. a** Daily alcohol intake (left) and average alcohol intake (right) of control and *Bmal1* heterozygote male mice. Two-way repeated measure ANOVA (RM-ANOVA), no significant effect, $F_{(1, 26)} = 2.793$, $p = 0.1067$, Unpaired two-tailed t-test, NS. **b** Daily alcohol intake (left) and average alcohol intake (right) of control and *Bmal1* heterozygote female mice. RM-ANOVA, no significant effect, $F_{(1, 25)} = 2.507$, $p = 0.1259$, Unpaired two-tailed t-test, NS. **c** Daily alcohol preference (left) and average alcohol preference (right) of control and *Bmal1* heterozygote male mice. RM-ANOVA, no significant effect, $F_{(1, 26)} = 1.326$, $p = 0.26$, Unpaired two-tailed t-test, NS. **d** Daily alcohol preference (left) and average alcohol preference (right) of control and *Bmal1* heterozygote female mice. RM-ANOVA, no significant effect, $F_{(1, 25)} = 3.506$, $p = 0.0729$, Unpaired two tailed t-test, NS. **e** Total fluid intake (left) and average fluid intake (right) of control and *Bmal1* heterozygote male mice. RM-ANOVA, no significant effect, $F_{(91, 26)} = 1.498$, $p = 0.2320$, Unpaired two-tailed t-test, NS. **f** Total fluid intake (left) and average fluid intake (right) of control and *Bmal1* heterozygote female mice. RM-ANOVA, no significant effect, $F_{(1, 25)} = 0.5874$, $p = 0.4506$, Unpaired two-tailed t-test, NS. **g** Daily alcohol intake (left) and average alcohol intake (right) of control and *Per2* heterozygote male mice. RM-ANOVA, no significant effect, $F_{(1, 13)} = 0.09317$, $p = 0.7650$. Unpaired two-tailed t-test, NS. **h** Daily alcohol intake (left) and average alcohol intake (right) of control and *Per2* heterozygote female mice. RM-ANOVA, no significant effect, $F_{(1, 13)} = 0.08137$. $p = 0.7799$. Unpaired two-tailed t-test, NS. **i** Daily alcohol preference (left) and average alcohol preference (right) of control and *Per2* heterozygote male mice. RM-ANOVA, no significant effect, $F_{(1, 13)} = 0.01314$, $p = 0.9105$. Unpaired two-tailed t-test, NS. **j** Daily alcohol preference (left) and average alcohol preference (right) of control and *Pe2* heterozygote female mice. RM-ANOVA, no significant effect, $F_{(1, 13)} = 0.04381$, $p = 0.8375$. Unpaired two-tailed t-test, NS. **k** Total fluid intake (left) and average fluid intake (right) of control and *Per2* heterozygote male mice. RM-ANOVA, no significant effect, $F_{(1, 13)} = 0.02780$, p = 0.8702. Unpaired two-tailed t-test, NS. **l** Total fluid intake (left) and average fluid intake (right) of control and *Per2* heterozygote female mice. RM-ANOVA, no significant effect, $F_{(1, 13)} = 1.833$, p = 0.1988. Unpaired two tailed t-test, NS. NS = no significant differences. The values express mean ± S.E.M. **a–f**, CTR: control, HET: *Bmal1* heterozygote. **a, c, e** CTR $n = 12$, HET $n = 16$. **b, d, f** CTR $n = 17$, HET $n = 10$. **g–l** CTR: control, HET: *Per2* heterozygote. **g, i, k** CTR $n = 8$, HET $n = 7$. **h, j, l** CTR $n = 9$, HET $n = 6$.

These results show that selective deletion of *Bmal1* from MSNs eliminates sex differences in alcohol intake and preference by augmenting consumption and preference in males and suppressing the heightened intake and preference in females. Likewise, striatal deletion of *Per2* eliminates sex differences in alcohol consumption, although the effect is attributed primarily to augmentation of intake and preference in Per2SKO males.

## Discussion
Our work reveals a novel role of *Bmal1* in striatal control of alcohol consumption. The influence of *Bmal1* is sexually dimorphic, associated with repression of alcohol consumption in male mice, potentially mediated by *Per2*, and enhancement of alcohol consumption in females via a mechanism independent of *Per2*. Both clock genes are associated with the regulation of alcohol intake in humans or animal models, but the exact mechanism of *Bmal1*- and *Per2*-dependent changes in alcohol consumption remains elusive. An increasing body of evidence suggest that alterations in glutamatergic signaling in the brain may contribute to the observed phenotypes[47–49].

Spanagel and colleagues observed elevated extracellular levels of glutamate due to an attenuated expression of transporter Eaat1 in Per2[Brdm1] mutant mice showing increased alcohol consumption[40].

Even though this mechanism is unlikely the cause for the phenotype observed in our study[50], it gives rise to the assumption that *Bmal1* and *Per2* affect striatal glutamatergic neurotransmission in a clock gene-specific and sex-dependent manner, conceivably by the neuromodulatory effects of gonadal hormones and/or dopamine. The expression of estrogen receptors has been shown to be clock controlled. Cai and colleagues demonstrated that the CLOCK-BMAL1 dimer binds to an E-box element in the estrogen receptor β (ERβ) promoter. The authors also showed that daily oscillations of ERβ expression, in both lung and skeletal muscle of WT mice, are abolished in BMAL1 knockout mice, indicating that ERβ is a direct target of the circadian clock[51], which may contribute to the sex- and clock gene-dependent effects on alcohol consumption in our mouse models. Interestingly, membrane-associated estrogen receptors activate metabotropic glutamate receptor signaling, thereby evoking sex-specific bidirectional responses via cAMP response element-binding protein (CREB) phosphorylation or dephosphorylation, respectively[52]. Even though alcohol consumption in mice appears to depend on phosphorylated CREB levels in the NAcc[53], it needs to be established in future studies whether an altered interaction of sex-hormone and glutamate signaling is directly connected to the observed alcohol drinking phenotypes in striatal clock gene knockout mice.

Alternatively, dopaminergic modulation of the glutamatergic neurotransmission may contribute to the observed differences in alcohol consumption in our mouse model[54]. It is conceivable that sex-specific factors may also play a role in the regulation of these processes, given that differences between males and females in striatal dopaminergic signaling are well established[55,56] and that dopamine receptor expression varies during the different stages of the estrous cycle. Furthermore, clock-dependent alterations in dopamine levels may contribute to the observed phenotypes. The expression of the monoamine oxidase A (MAOA), a key enzyme in dopamine degradation, is clock controlled[57–59]. Hampp and colleagues found reduced expression and activity of MAOA in the mesolimbic dopaminergic system, and increased levels of dopamine and altered neuronal activity in the striatum of *Per2* mutant mice[57]. Intriguingly, circadian clock genes and components of the dopamine signaling pathway in the striatum influence each other, adding to the complexity of the system. Treatment with DRD1 and DRD2 agonist evoked diverse expression of circadian clock genes in striatal neurons[60]. Because PER2 expression in MSNs is controlled by the DRD2 receptor[61], it is tempting to speculate that the observed clock gene-dependent variations in alcohol consumption in male and female mice may partially be explained by a cell-type specific mechanism. This is emphasized by a recent study revealing opposing roles of DRD1 and DRD2 MSNs in alcohol consumption by a selective activation of glutamatergic or GABAergic neurotransmission, respectively[62]. The exact role of circadian clock genes in a sex-, cell type- and region-specific neurotransmission in MSNs, and their impact on ethanol consumption, however, needs to be addressed in the future.

## Methods
**Animals and genotyping.** Conditional knockout mice lacking BMAL1 or PER2 protein in the striatum were generated by two genetic crosses. In a first cross, Gpr88(Cre/+) male mice (B6.129S4- Gpr88tm1.1(cre/GFP)Rpa/J; stock number 022510; Jackson Laboratory) were bred with Bmal1(fl/fl) (B6.129S4(Cg)-Arntltm1Weit/J; stock number 007668; Jackson Laboratory) or Per2(fl/fl) (B6.129-Per2tm1.2Ual/Biat, strain ID: EM10599, European Mouse Mutant Archive) female mice to generate respective heterozygote F1 progeny ([Gpr88Cre/+; Bmal1fl/+] or [Gpr88Cre/+; Per2fl/+]). In a second step, F1 males were crossed with Bmal1(fl/fl) or Per2(fl/fl) females to generate desired experimental and control animals. All floxed and Cre-expressing transgenic mouse lines have been backcrossed onto a C57BL/6 J background for at least 6 generations. Genomic DNA was isolated from tail biopsies[63], and genotyping was performed by polymerase chain reaction

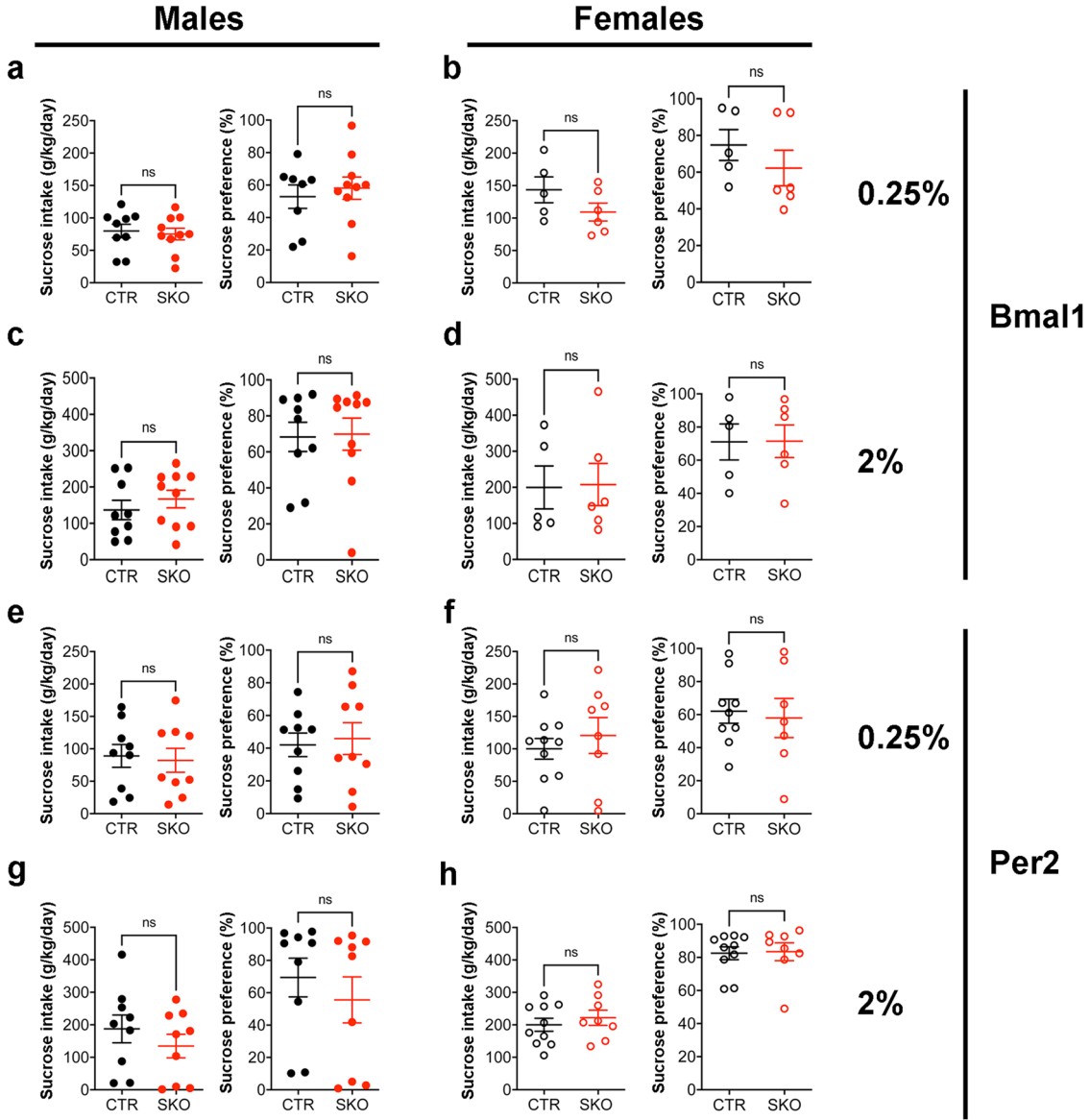

**Fig. 4 Drinking behavior of *Bmal1* and *Per2* knockout mice is independent of changes in the general reward processing. a** Average daily intake (left) and preference (right) of 0.25% sucrose solution of control and *Bmal1* knockout male mice. Unpaired two-tailed t-test, NS. **b** Average daily intake (left) and preference (right) of 0.25% sucrose solution of control and *Bmal1* knockout female mice. Unpaired two-tailed t-test, NS. **c** Average daily intake (left) and preference (right) of 2% sucrose solution of control and *Bmal1* knockout male mice. Unpaired two-tailed t-test, NS. **d** Average daily intake (left) and preference (right) of 2% sucrose solution of control and *Bmal1* knockout female mice. Unpaired two-tailed t-test, NS. **e** Average daily intake (left) and preference (right) of 0.25% sucrose solution of control and *Per2* knockout male mice. Unpaired two-tailed t-test, NS. **f** Average daily intake (left) and preference (right) of 0.25% sucrose solution of control and *Per2* knockout female mice. Unpaired two-tailed t-test, NS. **g** Average daily intake (left) and preference (right) of 2% sucrose solution of control and *Per2* knockout male mice. Unpaired two-tailed t-test, NS. **h**, Average daily intake (left) and preference (right) 2% sucrose solution of control and *Per2* knockout female mice. Unpaired two-tailed t-test, NS. NS = no significant differences. **a–h**, the values express mean ± S.E.M. **a–d**, CTR: *Bmal1* control, SKO: *Bmal1* Knockout. **a**, **c**, CTR n = 9, SKO n = 9. **b**, **d** CTR n = 5, SKO n = 6. **e–h** CTR: *Per2* control, SKO: *Per2* knockout. **e**, **g** CTR n = 9, SKO n = 9. **f**, **h** CTR n = 10, SKO n = 8.

(PCR) using the OneTaq® Hot Start 2X Master Mix (M0484, New England Biolabs Inc., Ipswich, MA, USA) according to the manufacturer's instructions. The specific primers for genotyping were: Cre: Fwd-5′-TTTTCACCTCCCTCCCTTCT-3′; Rev-5′- GCCCACGATTCTTCTTCCTC-3′, yielding a 255 bp product for wild type or a 180 bp product for mutant mice; Bmal1: Fwd-5′-CTGGAAG-TAACTTTATCAAACTG-3′, Rev-5′- CTGACCAACTTGCTAACAATTA-3′, yielding a 327 bp product for wild type or a 431 bp product for mutant animals; Per2: Fwd-5′-CTGTGTCCCTGGTTTCTG-3′, Rev-5′-GCAGGGCAGTTTCAT CAAGG- 3′, yielding a 468 bp product for wild type or a 596 bp product for mice carrying the transgene.

Mice were group-housed (2–4 individuals) under a 12:12 h light-dark cycle (08:00 am to 08:00 pm) with controlled temperature (22 ± 2 °C) and humidity

(40–60% relative humidity), and had access to water and standard rodent chow (5057, Charles River Laboratories, Wilmington, MA, USA) ad libitum. Conditional Bmal1 and Per2 knockout mice (Gpr88Cre/+; Bmal1fl/fl [Bmal1SKO], Gpr88Cre/+; Per2fl/fl [Per2SKO]), littermate heterozygote (Gpr88Cre/+; Bmal1fl/+ [Bmal1HET], Gpr88Cre/+; Per2fl/+ [Per2HET]) and wild type control animals (Gpr88+/+; Bmal1fl/fl [Bmal1CTR], Gpr88+/+; Per2fl/fl [Per2CTR]) of both sexes as well as Gpr88Cre/+ and corresponding control male and female mice were used for experiments with an age of 12–18 weeks. All animal experimental and animal care procedures in this report were conducted in accordance with the guidelines set forth by the Canadian Council of Animal Care and by the Animal Care Committee of Concordia University.

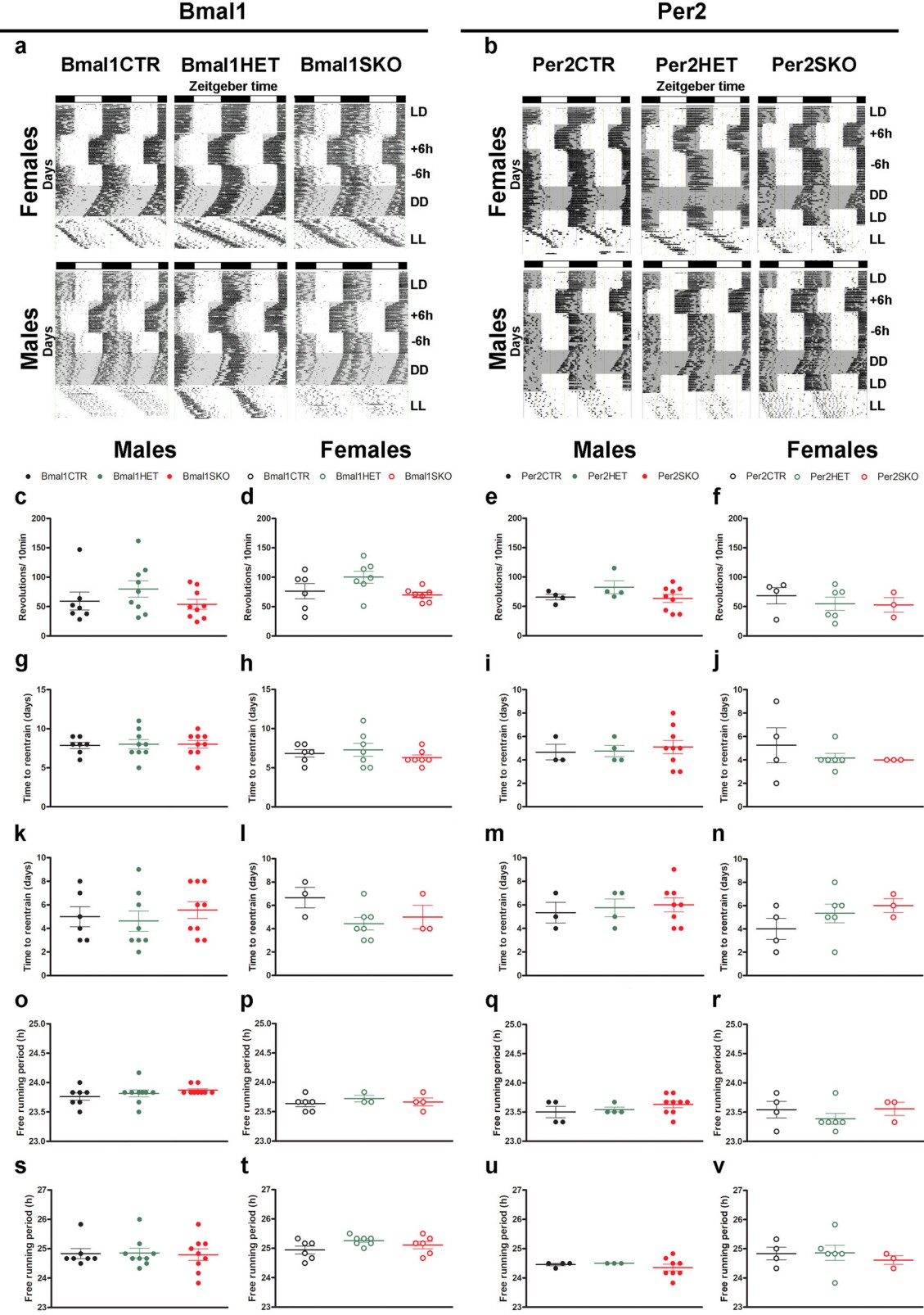

**Quantitative PCR**. Brains were isolated and flash-frozen from 11 to 13-week-old female Bmal1SKO, HET and CTL mice at ZT5, ZT11, or ZT17 (5, 11 and 17 h after lights turned on). 100 µm coronal section were obtained using a Microm HM 505 E Cryostat (Microm International, Walldorf, Germany) to collect brain tissue. RNA was extracted from tissue punches of the dorsal striatum (DS) and nucleus accumbens (NAcc) using the RNeasy Lipid Tissue Mini Kit (Qiagen, Hilden, Germany) and converted into cDNA by the iScript™ cDNA Synthesis Kit (Bio-Rad, Hercules, CA, USA) according to the manufacturer's instructions. qPCR was

performed using the following primer pairs: mBmal1: Fwd-5′-GCAGTGCC ACTGACTACCAAGA-3′, Rev-5′-TCCTGGACATTGCATTGCAT-3′; mPer2: Fwd-5′-TGATCGAGACGCCTGTGCTCG-3′, Rev-5′-CTCCACGGGGTTGATG AAGCTG-3′; mDbp: Fwd-5′-AAGAAGGCAAGGAAAGTCCAG-3′, Rev-5′-A CCTCTTGGCTGCTTCATTG-3′; mGapdh: Fwd- 5′-AGGTCGGTGTGAACGG ATTTG-3′, Rev-5′-TGTAGACCATGTAGTTGAGGTC-3′. EvaGreen based amplification detection (EVOlution master mix, MBI-E250, Montreal Biotech, Canada) was performed using an Eco™ real-time PCR system (Illumina, San Diego,

**Fig. 5 Conditional ablation of *Bmal1* and *Per2* does not affect SCN function. a** Representative double-plotted actograms illustrating the daily pattern of running-wheel activity of control, *Bmal1* heterozygote and knockout female (top), and male (bottom) mice. **b** Representative double-plotted actograms illustrating the daily pattern of running-wheel activity of control, *Per2* heterozygote and knockout female (top) and male (bottom) mice. The vertical marks indicate periods of activity of at least 10-wheel revolutions per 10 min. Each horizontal line plots 48 h, and sequential days are arranged from top to bottom. The empty and gray shaded areas in each actogram illustrate the light and dark phases, respectively. **c–v** Circadian analysis of locomotor activity of *Bmal1* and *Per2* control, heterozygote and knockout male and female mice. One way-ANOVA. No significant differences were observed between the different genotypes in any of the parameters analyzed. **c–f** Amplitude of the locomotor activity rhythm. **g–j**, time to entrain to a 6-h phase advance. **k–n** Time to entrain to a 6-h phase delay. **o–r** Free running period in constant dark (DD). **s–v** Free running period in constant light (LL). **a, b** LD, 12:12 h light dark; +6 h, 6-h phase advance; −6h, 6-h phase delay; DD, constant dark; LL, constant light. Bmal1CTR: control, Bmal1HET: *Bmal1* heterozygote, Bmal1SKO: *Bmal1* knockout. Per2CTR: control, Per2HET: *Per2* heterozygote, Per2SKO: *Per2* knockout. Bars on the graphs represent the arithmetic mean ± S.E.M.

CA, USA). Relative expression of Bmal1, Per2 and Dbp in relation to Gapdh was calculated based on the "delta delta Ct" ($\Delta\Delta$Ct) method[64] and normalized to the highest expression value across genotypes and time points for each gene.

**Western blot analysis.** 12–15 weeks old mice were adapted for 10 d to a 12:12 h light-dark cycle and sacrificed either 2 h (ZT2) or 14 h (ZT14) after lights turned on. Brains were isolated and flash-frozen in isopentane. A total of 200 µm coronal sections were cut on a cryostat (Microm HM 505 E Cryostat, Microm International, Walldorf, Germany). The striatal tissue was excised using punches and frozen on dry ice. Tissue was then homogenized in 150 µl of lysis buffer [1 M Tris-HCl pH 6.8, 10% sodium dodecyl sulfate, 0.1 ml phosphatase inhibitor cocktail 2 (Sigma, # P5726, Burlington, MA, USA), 0.1 ml phosphatase inhibitor cocktail 3 (Millipore Sigma, P0044, Burlington, MA, USA), 1 × protease inhibitor cocktail (Roche, Basel, Switzerland)]. Protein extracts (40 µg/lane) were electrophoresed into a 10% SDS-PAGE gel and then transblotted onto a nitrocellulose membrane (Bio-Rad, # 1620112, Hercules, CA, USA). Membranes were blocked in 5% Milk TBST buffer (skim milk powder) and then incubated (overnight, 4 °C) in TBST (with 5% skim milk powder) with the anti-Bmal1 (1:1000 dilution) antibody (Novus Biologicals, # NB100-2288, Littleton, CO, USA). The same antibody was used for immunolabeling analysis. Next, the membrane was incubated in TBST (with 5% milk) with a goat anti-rabbit IgG horseradish peroxidase-conjugated antibodies (1:200 dilution; Millipore Sigma, # AP132P, Burlington, MA, USA). The signal was visualized using the Western Lighting Chemiluminescence light-emitting system (PerkinElmer Life Sciences, Waltham, MA, USA). Between each antibody treatment, membranes were washed a minimum of three times (10 min per wash) in TBST.

**Tissue preparation for Immunohistochemistry.** 12–15 weeks old mice were transcardially perfused by cold saline (0.9% NaCl), followed by cold paraformaldehyde solution (4% in a 0.1 M phosphate buffer, pH 7.3). Brains were extracted and stored overnight in 4% paraformaldehyde solution at 4 °C. Coronal sections were collected using a Leica vibratome, sliced at a thickness of 50 µm (immunohistochemistry) or 30 µm (immunofluorescence) and then stored at −20 °C in Watson's cryoprotectant[65] until further use.

**Immunohistochemistry.** Free-floating sections were rinsed once for 10 min in phosphate buffered saline (PBS, pH 7.4), followed by 3 × 10 min rinses in 0.3% Triton-X in Trizma-buffered saline solution (TBST: 0.3% Triton, 50 mM Trizma buffer, 0.9% saline). Immunohistochemistry for BMAL1 was performed using an affinity-purified rabbit polyclonal antibody, raised against BMAL1 (1:10000 Novus Biologicals # NB100-2288, Littleton, CO, USA). Brain sections were incubated (40 h, 4 °C) in a primary solution with Bmal1 polyclonal rabbit antibody, 2% normal goat serum, 5% milk buffer in TBST. The sections were then incubated in a secondary solution, composed of biotinylated anti-rabbit IgG, raised in goat (1:200, Vector Laboratories, Burlington, ON, Canada). Lastly, the sections were incubated in a tertiary Avidin-Biotin-Peroxidase solution (Vectastain Elite ABC Kit, Vector Laboratories, Burlington, ON, Canada). All sections were rinsed in a 0.5% 3,3- diaminobenzidine (DAB) solution. Immunoreactive (IR) cells were stained using a 0.5% DAB, 0.01% $H_2O_2$ and 8% $NiCl_2$ solution. Stained sections were mounted onto gel-coated slides, dehydrated in a series of alcohols and Citrisolv (Fisher Scientific, Pittsburgh, PA, USA), covered by Permount media (Fisher Scientific, Pittsburgh, PA, USA), and coverslipped. The sections were examined under a light microscope (Leica, DMR) and identified using Paxinos mouse brain atlas[66].

Images of the SCN, striatum, and hippocampus were captured using a Sony XC-77 video camera, Scion LG-3 frame grabber (Scion Corporation, Frederick, MD, USA), and Image J[67].

**Immunofluorescence.** Free-floating sections were rinsed once for 10 min in phosphate buffered saline (PBS, pH 7.4), followed by 3 × 10 min rinses in 0.3%

Triton-X in PBS (PBST). Tissue was pre-blocked for 1 h at room temperature with gentle agitation in a solution of PBST containing 3% skim milk powder and 6% normal donkey serum (NDS) then directly transferred to the primary antibody incubation. The tissue was incubated for 48 h with the primary antibody at 4 °C with gentle agitation, rinsed 3 × 10 min in PBST, then incubated with the secondary antibody for 1 h at room temperature with gentle agitation. Antibodies were diluted in a solution of 0.3% PBST with 3% skim milk powder and 2% NDS. The following antibodies and dilutions were used: PER2 rabbit polyclonal (1:500, Novus Biologicals, # NB100-125, Littleton, CO, USA), BMAL1 rabbit polyclonal (1:500, Novus Biologicals # NB100-2288, Littleton, CO, USA), anti-rabbit secondary Alexa-647 (1:500, Life Technologies, Carlsbad, CA, USA). Once all incubations were complete, the tissue was rinsed 3 × 10 min in PBST and 10 min in PBS before being mounted onto slides, allowed to air dry and coverslipped with VECTASHIELD® Antifade Mounting Media (Vector Laboratories, Inc., Burlingame, CA, USA). Slides were left to cure overnight in the dark, sealed with clear nail polish and imaged over the next 5 days. Slides were stored in a slide box at 4 °C. Fluorescent images were captured using the 60 × objective on an Olympus FV10i automated confocal laser scanning microscope at the Centre for Microscopy and Cell Imaging, Concordia University, Montreal, Canada. Brain regions of interest were determined using Paxinos mouse brain atlas[66]. All confocal parameters (pinhole, contrast, brightness, etc.) were held constant across all data sets from the same experiment.

**Two-bottle choice- intermittent access.** 12–18 weeks old male and female mice ($n = 6$–17) were separated from group housing one week prior to the beginning of the experiment and single-housed under a 12:12 h light-dark cycle. Oral alcohol intake was determined using 24-h intermittent access to alcohol in a two-bottle choice drinking paradigm. Every other day, at ZT4 (4 h after lights were on), they were given 24 h of concurrent access to one bottle containing 15% ethanol (v/v) in tap water and one bottle containing tap water for 11 sessions of alcohol access. The bottles were weighed every day before and after giving them to the mice. The mice were weighed at the beginning and once a week during the experiment. The position (left or right) of each solution was alternated between sessions as a control for side preference. The possible loss of solutions due to the handling of the bottles, was controlled by weighing bottles in empty cages.

**Sucrose consumption.** 12–18 weeks old male and female mice ($n = 5$–10) were single-housed under a 12:12 h light-dark cycle. Every day, at ZT4, they were given free access to one bottle containing 0.25% sucrose in tap water (m/v) and 1 bottle containing tap water, in alternate left-right position.

Sucrose solution was offered for 3 consecutive days. The amount of fluid intake was recorded every day before and after giving them to the mice. One week later, the same procedure was performed with 2% sucrose solution. The bottles were weighed every day and the mice were weighed at the beginning of the experiment.

**Locomotor activity rhythms.** 12–15 weeks old male and female mice were housed individually in cages equipped with a running wheel with ad libitum access to food and water. Each cage was kept in a temperature-controlled, soundproof, ventilated isolation chamber and with computer-controlled light sources. Main light source was adjusted such that approximately 100–300 lux of light was equally distributed in the chamber. Mice were kept under a 12:12 h light-dark (LD) schedule, and thereafter exposed to 6 h phase advance (+6 h), followed by a 6 h phase delay (−6 h). After that, mice were transferred into constant darkness (DD) followed by exposure to constant light (LL). Mice were kept for 21–40 days under the respective light regimen. Wheel rotation was recorded using the VitalView program (Mini Mitter, Starr Life Sciences Corp., Oakmont, PA, USA) and double-plotted actograms were prepared using the Actiview software (Mini Mitter, Starr Life Sciences Corp., Oakmont, PA, USA). Microsoft Excel and ClockLab 6 (Actimetrics, Wilmette, IL, USA) were used to analyze locomotor activity patterns. Amplitude was analyzed from summed activity counts recorded over 10 min intervals over a period of 7–10 days. Time to re-entrain to 6-h phase shifts was determined for activity onset and offset

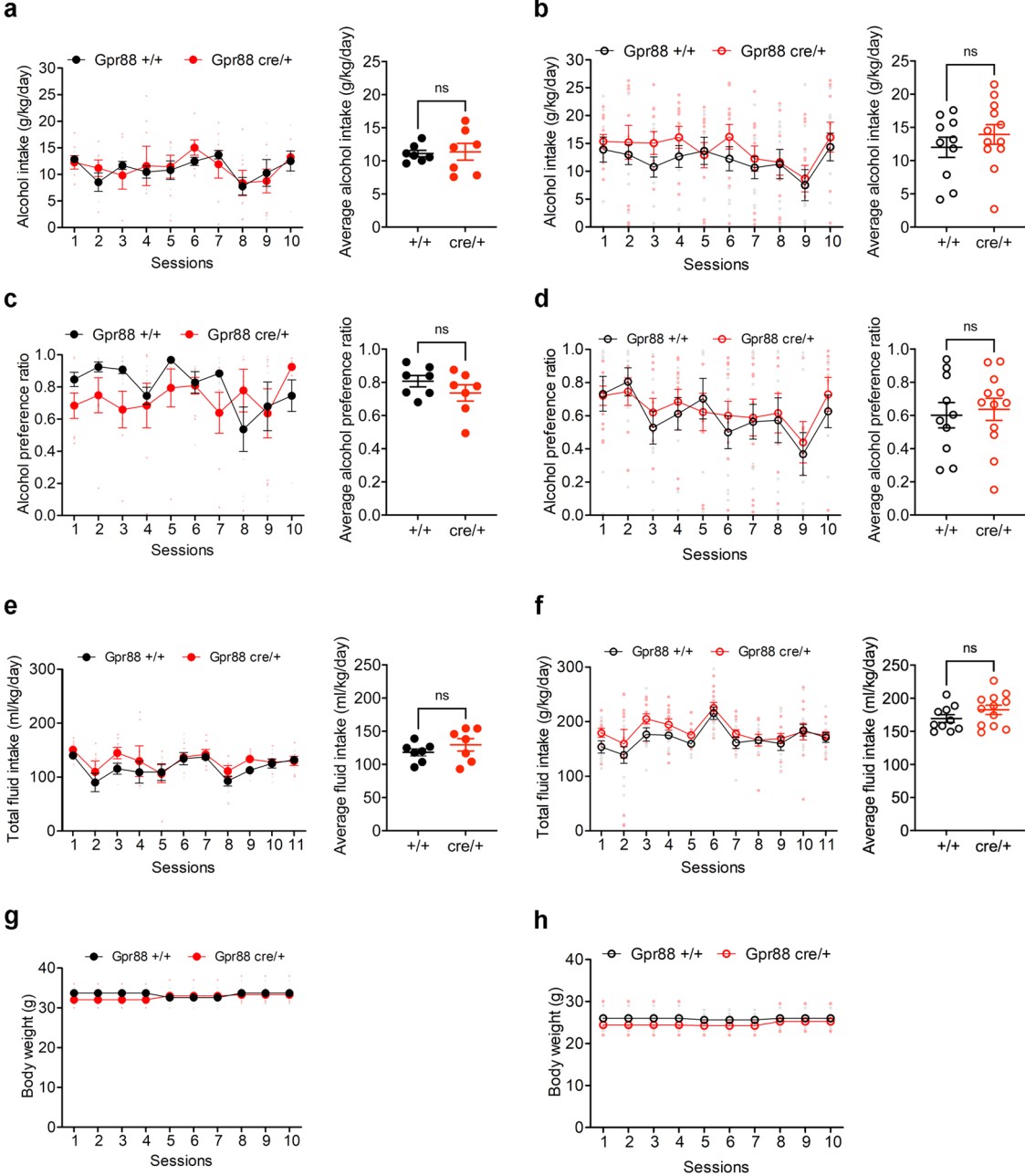

**Fig. 6 Alcohol consumption is not altered in Gpr88 heterozygote mice. a** Daily alcohol intake (left) and average alcohol intake (right) of control and *Gpr88* heterozygote male mice. Two-way repeated measure ANOVA (RM-ANOVA), no significant effect, F (1, 12) = 0.03709, *p* = 0.8505. Unpaired two-tailed t-test, no significance. **b** Daily alcohol intake (left) and average alcohol intake (right) of control and *Gpr88* heterozygote female mice. RM- ANOVA, no significance, F (1, 20) = 0.8286, *p* = 0.3735. Unpaired two-tailed t-test, no significance. **c** Daily alcohol preference (left) and average alcohol preference of control and *Gpr88* heterozygote male mice. RM-ANOVA, no significance, F (1, 12) = 1.418, *p* = 0.2567. Unpaired two-tailed t-test, no significance. **d** Daily alcohol preference (left) and average alcohol preference of control and *Gpr88* heterozygote female mice. RM-ANOVA, no significance, F (1, 20) = 0.1233, *p* = 0.0.7292. Unpaired two-tailed t-test, no significance. **e** Total fluid intake (left) and average fluid intake (right) of control and *Gpr88* heterozygote male mice. RM-ANOVA, no significant effect, F (1, 12) = 1.079, *p* = 0.3195, Unpaired two-tailed t-test, NS. **f** Total fluid intake (left) and average fluid intake (right) of control and *Gpr88* heterozygote female mice. RM-ANOVA, no significant effect, F (1, 20) = 1.983, *p* = 0.1744, Unpaired two-tailed t-test, NS. **g** Daily body weight of control and *Gpr88* heterozygote male mice. RM-ANOVA, no significant effect, F (1, 12) = 0.5672, *p* = 0.4659. **h** Daily body weight of control and *Gpr88* heterozygote female mice. RM-ANOVA, no significant effect, F (1, 20) = 2.905, *p* = 0.1038. NS = no significant differences. The values express mean ± S.E.M. **a**, **c**, **e**, **g**, Gpr88+/+, *n* = 7 and Gpr88Cre/+, n = 7. **b**, **d**, **f**, **h** Gpr88+/+, *n* = 10 and Gpr88Cre/+, *n* = 12.

individually. The criterion point was a stable alignment of the onset and offset to the shifted light:dark regimen, and the difference between the day at the shift, and the first day of the re-entrained activity phase was calculated. The ActogramJ plugin[68] of the Fiji software package[69] was used to determine the free-running period in constant darkness and constant light.

**Statistics and reproducibility.** Statistical analysis was conducted using the GraphPad Prism 8 software package (GraphPad software LLC, San Diego, CA, USA). Unpaired two-tailed t test and ANOVAs with or without repeated measures have been used to compare differences between groups. Significant main effects and interactions of the ANOVAs were further investigated by the appropriate

post-hoc test. The level of statistical significance was set at $P < 0.05$. Exact $P$ values and further details are given in the text.

**Reporting summary**. Further information on research design is available in the Nature Research Reporting Summary linked to this article.

## Data availability

The datasets generated during and/or analysed during the current study are available from the corresponding author on reasonable request.

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

## Acknowledgements

This work was funded by grants from the Canadian Institutes of Health Research (S.A). All microscopy was performed at the Concordia Centre for Microscopy and Cellular Imaging at Concordia University, Montreal (special thanks to Dr. Chris Law). Instrumentation for quantitative PCR was provided by the Centre for Structural and Functional Genomics at Concordia University, Montreal (special thanks to Dr. Marc Champagne)

## Author contributions

S.A. and N.Z. conceived and designed the study; N.Z., P.S., S.F., J.G., G.P. performed the experiments; S.A., K.S. N.Z. analyzed and Interpreted the data, and wrote the manuscript; S.A. and N.Z. Supervision, N.Z. Project Administration.

## Competing interests

The authors declare no competing interests.
