## [Peer Review File · Communications Biology]

Reviewers' comments:

Reviewer #1 (Remarks to the Author):

The authors show that lack of Bmal1 in the striatum influences alcohol intake in mice in a sexually dimorphic manner. Interestingly, lack of Per2 in the striatum only affects alcohol intake in males but not females. The study is well controlled except for one thing the authors may be able to provide easily. The specific deletion of Bmal1 or Per2 in the striatum is shown by immunohistochemistry, but the authors do not show other brain regions such as the SCN and Cortex expressing Bmal1 or Per2 normally (control). This would strengthen the paper.

Minor comments:

- 1) page 3, line 4 from bottom of the page: two times that, remove one
- 2) In the text citation of numerous figures are not correct, please adjust:
page 5, middle paragraph, third line from below: (Figs 3a, c, e, g) and (Figs. 3b, d, f, h)
page 6, third line from top: (Fig. 3i, j).
- page 7, second line 4a
fourth line, 4b
fifth line, 4c, d
seventh line, 4c, d

Reviewer #2 (Remarks to the Author):

Summary:

This manuscript by Zavalia et al is focused on determining a role for the circadian clock gene, Bmal1, in alcohol consumption. Previous studies have implicated a role for Bmal1 in alcohol drinking behavior, but a causal role for Bmal1 and the potential cell types through which it may influence drinking have not been investigated. Here, the authors utilized male and female mice that lack Bmal1 in medium spiny neurons (MSNs) in the striatum to determine a causal role for Bmal1 in these neurons in alcohol drinking and preference. They showed that Bmal1 knockdown in MSNs produces a sexually dimorphic effect on alcohol drinking and preference, with an increase in preference and intake in males and a decrease in females. Furthermore, MSN-specific deletion of Per2 in the striatum also increased alcohol intake and preference in males (similar to Bmal1), but had no effect in females, suggesting a potential Per2-dependent mechanism in males. Overall, the manuscript is well-written and the data implicate a novel, sex-specific role for Bmal1 in alcohol consumption and preference. I have some minor issues that the authors should address:

Comments:

1. It would be helpful to include more information in the abstract, introduction, and discussion to put the findings in context. For example:
 - a. Abstract – The Per2 results are very interesting but not mentioned in the abstract
 - b. Introduction – The introduction is very short. It would be helpful to expand upon previous studies regarding the roles of sex, circadian rhythms, and MSNs in alcohol drinking to provide rationale for the current study.
 - c. Discussion – Expand the discussion on potential mechanisms for the sex-specific findings (e.g., Per2 dependent/independent mechanisms in males vs females).
2. Quantification of some of the results are missing in the manuscript:
 - a. The western blot results in Extended Data Fig. 1a should indicate the molecular weight and should be quantified to show knockdown.
 - b. The activity rhythms in Fig. 3i and 3j should be quantified.
3. Are the ANOVAs in Figures 1 and 2 (alcohol consumption and preference) repeated measures ANOVAs? If so, this should be clarified.
4. In Extended Data Fig. 1b, it would be helpful to show the lack of expression in the striatum for comparison.
5. Pg. 5, Lines 102-104 – the authors state that there is "...no difference in total liquid consumption..", but there is a difference in total liquid consumption in females (Fig. 2h). Could this be influencing the lack of effect in female mice?

Minor Points:

1. It would be helpful to split the manuscript up into sections with headers.
2. Pg. 2, Line 25 – “Bone” should be “Brain”
3. Pg. 6, Line 124 – “Fig. 3e, f” should be “Fig. 3i, j”
4. Pg. 7, Lines 150-151 – “Extended Data Fig. 4c, f” should be “4a, b”. “Extended Data Fig. 4g, J” should be “4c, d”

Reviewer #3 (Remarks to the Author):

Summary of the manuscript

The authors investigate the influence of striatal BMAL1 and PER2, on the gender differences in the propensity to consume alcohol in mice. They report that the selective deletion of *Bmal1* and *Per2* in medium spiny neurons increased voluntary alcohol intake in males, while *Bmal1*, but not *Per2*, repressed the intake in females.

Overall impression of the work

This is a novel and interesting study. The authors provide strong data indicating that the expression of BMAL1 in striatal medium spiny neurons plays a critical role in controlling dimorphic alcohol consumption in mice. They also investigate the implications of PER2 in these sexually dimorphic phenotypes, by deleting *Per2* from the medium spiny neurons, in male and female mice. They found that while PER2 plays a role in the increment of alcohol intake and preference in male mice, it did not affect alcohol consumption and preference in females.

The manuscript is very clear and precise. The statistical analysis is appropriated, and the experimental procedures are detailed and reported with sufficient precision to allow the reproducibility of the work.

I have one major and several minor comments that are indicated below.

Major comment:

Concerning the differences in the PER2-control of alcohol intake in males versus females, is *Per2* oscillating in the striatum of both males and females? Moreover, does the deletion of *Per2* in the MSNs alter the oscillatory expression of BMAL1 or the striatal clock in both males and females? Would be interesting to check the circadian expression of *Per2* and other clock genes (as in Figure 1, such as *Dbp* but also *Cry1*, etc) in both males and females to verify whether the action of *Bmal1* and/or *Per2* in the control of alcohol intake underly a circadian function and/or involved the striatal clock.

Minor comments:

1. In figure 1a, are *Bmal1*, *Per2*, and *Dbp* oscillating in the striatum in both males and females?. From the western blots in the extended data Fig1a, it doesn't seem there are differences in the expression in BMAL1 at ZT2 vs ZT14. Moreover, in Fig1b, it is not specified where the expression levels of *Per2* and *Dbp* are analyzed in males or females. Please, provide mRNA or protein expression levels of *Bmal1*, *Per2*, and *Dbp* in the striatum of males and females at different ZTs.
2. In figure 1b, it seems there is a loss of BMAL1-positive cells among the GFP-negative population in *Bmal1*-STO versus *Bma1*-HET. What is the proportion of BMAL+ cells/GFP negative and/or positive in *Bmal1*-STO and *Bma1*-HET?. Please provide quantification of *Bmal1*-, GFP- positive cells and their colocalization.
3. Please, provide higher quality images for the Figure 2a and b as well as the quantifications of the GFP and/or PER2/BMAL1 and specify the ZT at which the animals were harvested.
4. From the representative actogram of the Females *Per2*-CTR in constant darkness, seems that

this animal had a very different period than the Per2HET or Per2STO. Please, provide the free-running period of the Bmal1 and Per2 animals (males and females).

Answers to reviewers:

Reviewer # 1	
Comment	Answer
1. The authors do not show other brain regions such as the SCN and Cortex expressing Bmal1 or Per2 normally (control). This would strengthen the paper.	Thank you for the comment. Ext. Data Fig 1b shows the expression of Bmal1 in the SCN and hippocampus of control and Bmal1 knockout mice. Cortex sections of PER2 immunofluorescence have being added to Ext. Data Fig 1.
2. Page 3, line 4 from bottom of the page: two times that, remove one	Thank you. Rectified on the manuscript. Page 6, line 70
3. In the text citation of numerous figures are not correct, please adjust: page 5, middle paragraph, third line from below: (Figs 3a, c, e, g) and (Figs. 3b, d ,f, h) page 6, third line from top: (Fig. 3i, J). page 7, second line 4a fourth line, 4b fifth line, 4c, d seventh line, 4c, d	Thank you. Adjusted on the manuscript
Reviewer # 2	
1. It would be helpful to include more information in the abstract, introduction, and discussion to put the findings in context. For example: a. Abstract – The Per2 results are very interesting but not mentioned in the abstract b. Introduction – The introduction is very short. It would be helpful to expand upon previous studies regarding the roles of sex, circadian rhythms, and MSNs in alcohol drinking to provide rationale for the current study. c. Discussion – Expand the discussion on potential mechanisms for the sex-specific findings (e.g., Per2 dependent/independent mechanisms in males vs females).	Thank you. We agree, and we have added more information. 1.a. Modified on the manuscript 1.b. Modified on the manuscript 1.c. Modified on the manuscript
2. Quantification of some of the results are missing in the manuscript: a. The western blot results in Extended Data Fig. 1a should indicate the molecular weight and should be quantified to show knockdown. b. The activity rhythms in Fig. 3i and 3j should be quantified.	Thank you for the comment. We have made modifications. a. Modified on the manuscript. b. Data added to the manuscript (see below in response to Reviewer 3).
3. Are the ANOVAs in Figures 1 and 2 (alcohol consumption and preference) repeated	Good question. Rectified on the manuscript

measures ANOVAs? If so, this should be clarified.	
4. In Extended Data Fig. 1b, it would be helpful to show the lack of expression in the striatum for comparison.	We agree, and we've added the staining of the striatum.
5. Pg. 5, Lines 102-104 – the authors state that there is “...no difference in total liquid consumption..”, but there is a difference in total liquid consumption in females (Fig. 2h). Could this be influencing the lack of effect in female mice?	Thank you very much for this comment. We thought about this question, but even though total fluid consumption is significantly higher in Per2SKO than in the PER2CTR females, we cannot hypothesize that the increase in total fluid intake influences the lack of effect. Alcohol intake does not change between genotypes, so if total fluid intake is higher in the Per2SKO females, we would've expected to see an effect on alcohol preference. We think that the increase in total fluid intake is not big enough to produce a significant effect since we do not see changes in alcohol preference between the two genotypes. But, further studies have to be performed to answer this question.
6. It would be helpful to split the manuscript up into sections with headers.	We agree, and we adjusted the manuscript
7. Pg. 2, Line 25 – “Bone” should be “Brain”	Thank you. Rectified on the manuscript. Page 2, line 31.
8. Pg. 6, Line 124 – “Fig. 3e, f” should be “Fig. 3i, j”	Thank you. Rectified on the manuscript.
9. Pg. 7, Lines 150-151 – “Extended Data Fig. 4c, f” should be “4a, b”. Extended Data Fig. 4g, J” should be “4c, d	Thank you. Rectified on the manuscript
Reviewer # 3	
1. Concerning the differences in the PER2-control of alcohol intake in males versus females, is Per2 oscillating in the striatum of both males and females?	Per2 expression varies over the course of a day in Bmal1 control animals (see Fig 1), and we expected that it does in Per2 control animals of both sexes too. The differences in alcohol intake and preference between males and females observed in the Per2 control mice are similar to the differences seen in Bmal1 control mice and according to what has been shown previously by other authors in alcohol intake in WT mice. Unfortunately, due to Covid-19 and the restrictions imposed to contain the spread of the pandemic, we are not working at full capacity, and the colony has been extremely reduced. For this reason, we cannot corroborate the oscillation of Per2 in the striatum of both males and females of our Per2 line. We have previously shown that PER2

	oscillates in the striatum of male rats (Harbour VL, Weigl Y, Robinson B, Amir S. Comprehensive mapping of regional expression of the clock protein PERIOD2 in rat forebrain across the 24-h day. PLoS One. 2013 Oct 4;8(10):e76391. doi: 10.1371/journal.pone.0076391). The phase and amplitude of the oscillation vary within the different subregions of the striatum (dorsal and medial, nucleus accumbens core, nucleus accumbens shell). The striatum is a sexually dimorphic brain region; thus, sex differences might occur in the phase and amplitude of the oscillation in the different striatal subregions but will be investigated in the future.
2. Does the deletion of Per2 in the MSNs alter the oscillatory expression of BMAL1 or the striatal clock in both males and females?	Preliminary immunostaining data from our lab have shown BMAL1 expression in the striatum of both males and females (Figure 2b). Both genes are vital components of the transcription-translation feedback loops (TTFL), and they regulate each other. Bmal1 expression is also regulated by another feedback loop involving RORs and REV-ERBα. Future studies will determine if and how the deletion of Per2 affects the expression of Bmal1 in a sex-dependent manner.
3. Would be interesting to check the circadian expression of Per2 and other clock genes (as in Figure 1, such as Dbp but also Cry1, etc) in both males and females to verify whether the action of Bmal1 and/or Per2 in the control of alcohol intake underlies a circadian function and/or involved the striatal clock.	Preliminary data from our lab show alterations in the expression of mRNA of Cry1 and Cry2 in Bmal1 knockout male mice during the 24 h day. Further studies will be carried out to determine the rhythms of mRNA expression of clock and clock-controlled genes in males and females. Unfortunately, due to the Covid-19 pandemic restrictions, we don't have the tools and resources to perform these experiments at the moment. Future experiments will be planned to address this question.
4. In figure 1a, are Bmal1, Per2, and Dbp oscillating in the striatum in both males and females?	The oscillation analysis has been made only in male mice. Future experiments will determine the pattern in females and whether sex differences exist.
5. From the western blots in the extended data Fig1a, it doesn't seem there are differences in the expression in BMAL1 at ZT2 vs ZT14. Moreover, in Fig1b, it is not specified where the expression levels of Per2 and Dbp are analyzed in males or females. Please, provide mRNA or protein expression levels of Bmal1, Per2, and Dbp in the striatum of males and	The aim of our work was to study the effect of knocking out the Bmal1 or Per2 gene on alcohol drinking. We successfully knocked out both genes from the striatum, as shown in Figures 1 and 2. The western blot is a qualitative experiment to confirm the downregulation in the expression of Bmal1 seen by immunofluorescence and qPCR. The number of

females at different ZTs.	animals used in this experiment is not enough to properly quantify the amount of protein. In the images, it is clear that there is less BMAL1 in the Bmal1 knockout mice. Since these genes are core clock genes involved in the generation and maintenance of circadian rhythms, we measure the expression of BMAL1 at two different time points to show that there is a decrease of expression at different times of the day. To determine if there is a rhythm of expression and analyze the phase and amplitude of that rhythm, it is necessary to measure the protein levels at least at 4 time points. The lack of difference in protein level between 2 time points does not mean there is no rhythm because the phase of the rhythm could have been shifted. It is also vital to take into consideration the individual's variability in protein expression. To determine if there is a rhythm of expression, it is recommended to pool the sample of multiple animals to have a more uniform representation of each time point.
6. In figure 1b, it seems there is a loss of BMAL1-positive cells among the GFP-negative population in Bmal1-STO versus Bma1-HET. What is the proportion of BMAL+ cells/GFP negative and/or positive in Bmal1-STO and Bma1-HET?. Please provide quantification of Bmal1-, GFP- positive cells and their colocalization.	The aim of this paper is paper was to study the involvement of striatal Bmal1 in the regulation of alcohol drinking. To do so, we ablated Bmal1 from MSNs, which represent 95% of the neuron population in the striatum. We show a significant decrease of Bmal1 expression by qPCR, immunofluorescence and western blot in our knockout model. Since we haven't performed Dapi staining, we cannot quantify cells that are both negative for GFP and Bmal1, so we will be misestimating the proportion of each type of cell. Unfortunately, due to Covid-19 restrictions, we cannot redo the staining in new samples. Quantifying the cells would provide. Further studies need to be performed to determine the contribution of the non-MSN cell to the observed phenotype.
7. Please, provide higher quality images for the Figure 2a and b as well as the quantifications of the GFP and/or PER2/BMAL1 and specify the ZT at which the animals were harvested.	Modified on the legend of the figure.
8. From the representative actogram of the Females Per2-CTR in constant darkness, seems that this animal had a very different period than the Per2HET or Per2STO. Please,	Thank you for pointing this out. Modifications have been made to the manuscript, and data has been added.

provide the free-running period of the Bmal1 and Per2 animals (males and females).	
--	--

REVIEWERS' COMMENTS:

Reviewer #1 (Remarks to the Author):

The authors have answered my questions and I have no further comments.

Reviewer #2 (Remarks to the Author):

The authors have addressed the concerns from the prior review.

Reviewer #3 (Remarks to the Author):

My congratulations to the authors. The authors have clarified a number of points and my initial concerns have been addressed.